# Role of Magnetic Resonance Imaging in Pelvic Organ Prolapse Evaluation

**DOI:** 10.3390/medicina59122074

**Published:** 2023-11-24

**Authors:** Giuseppe Sarpietro, Pietro Valerio Foti, Carmine Conte, Maria Grazia Matarazzo

**Affiliations:** 1Gynecological and Obstetrics Unit, Department of General Surgery and Medical Surgical Specialties, University of Catania, Via Santa Sofia 78, 95123 Catania, Italy; 2Radiodiagnostic and Radiotherapy Unit, University Hospital “Policlinico-San Marco”, 95123 Catania, Italy; pietrofoti@hotmail.com; 3Department of Woman and Child Health and Public Health, Catholic University of the Sacred Heart, 00168 Rome, Italy; carmine.conte87@gmail.com; 4Department of Obstetrics and Gynaecology, Azienda di Rilievo Nazionale e di Alta Specializzazione (ARNAS) Garibaldi Nesima, 95124 Catania, Italy; mariagraziamatarazzo@gmail.com

**Keywords:** pelvic organ prolapse, magnetic resonance imaging, vaginal surgery

## Abstract

*Background and Objectives*: The primary method for assessing pelvic floor defects is through physical examination. Magnetic resonance imaging (MRI) is a radiological technique that is useful for identifying the underlying defects of pelvic floor structures that require surgery. The primary aim of this study was to find correspondence between the clinical and radiological staging of pelvic organ prolapse (POP) before and after vaginal surgery. A secondary endpoint was to investigate, through clinical and MRI findings, whether surgery influences continence mechanisms. Finally, we reported changes in the quality of life of women who underwent surgery for prolapse. *Materials and Methods*: Twenty-five women with prolapse stage ≥ 2 POP-Q were recruited in this prospective study. They underwent preoperative clinical examination, MRI at rest and under the Valsalva maneuver, and quality of life questionnaires. Three months after vaginal surgery, they repeated clinical and radiological evaluation. *Results*: Twenty women completed the study. Both clinical and MRI evaluations showed an improvement in prolapse and symptoms after surgery. There were some discrepancies between clinical and radiological staging. MRI parameters did not show differences between pre- and postoperative values at rest; under the Valsalva maneuver, instead, the measurements changed after surgery. Continence was not worsened by the widening of the vesicourethral angle. Patients reported an improvement in quality of life. *Conclusions*: MRI is an accurate and objective method for defining the stage of prolapse, but clinical evaluation alone is sufficient for staging prolapse before surgery and evaluating the result at follow-up. It is an accurate method for visualizing some pelvic structures that can be compromised because of pelvic organ prolapse. MRI showed that vaginal surgery does not affect continence mechanisms.

## 1. Introduction

Pelvic organ prolapse (POP) is a common disease adversely affecting the quality of life of many women [1]. It affects 50% of parous women [2] and 37% of women over the age of 80 [3]. The lifetime probability of undergoing surgery for pelvic organ prolapse is 11% [4]. POP is defined as the descent of one or more of the anterior vaginal wall, the posterior vaginal wall, the uterus (cervix), or the apex of the vagina (vaginal vault or cuff scar after hysterectomy) [5]. It has been attributed to both damage to the levator ani muscle [6], where weakness may cause widening of the levator hiatus and descent of the central portion of the pelvic diaphragm, as well as to endopelvic fascial defects [7]. However, DeLancey described the interaction between pelvic floor muscles and the endopelvic fascia and maintained that determining which of these factors is solely responsible for prolapse is challenging as they are intimately interdependent [8,9].

Clinical investigation of the female pelvic floor has advanced significantly in recent years, and nowadays prolapse can be staged from I to IV through the Pelvic Organ Prolapse Quantification (POP-Q) introduced in 1996 [10,11,12]. While a thorough physical examination is a primary method for assessing pelvic floor defects, even the most experienced clinicians can occasionally be confounded by multiple findings [13] and by many variables (time of the day in which the clinician makes the examination, the position of the examination, etc.), and this could compromise the success of the treatment. To gain greater certainty, clinicians have a wide range of imaging modalities at their disposal, which can confirm clinical suspicions and uncover unexpected defects. Despite being an expensive radiological technique, magnetic resonance imaging (MRI) offers well-known advantages, including the absence of radiation exposure, excellent soft tissue contrast, and the ability to provide multiplanar imaging without superimposition of structures. These advantages make it possible to visualize the morphology of the pelvic floor in great detail [14]. Functional MRI (fMRI) of the pelvic floor, depicting organ movement, was first introduced by Yang et al. and Kruyt et al. in 1991 [15]. Radiologists can accurately identify and report the underlying structural defects, enabling clinicians to customize surgical techniques for each patient. This is crucial because even patients with similar clinical symptoms may have varying underlying structural derangements or abnormalities, necessitating distinct treatment plans or approaches. The primary endpoint of our study was to find a correspondence between the clinical and radiological classification of pelvic organ prolapse before and after vaginal surgery. A secondary objective was to demonstrate any correlation between vaginal surgery and continence, through analysis of some pelvic measurements in MRI.

## 2. Materials and Methods

This study was performed at the Urogynecological Service of Gynaecological Clinic and Radiodiagnostic and Radiotherapy Unit, University of Catania, Italy. It was a prospective study including 25 women recruited who underwent surgery for urogenital prolapse between October 2018 and February 2019. The institutional ethical committee of the department approved the study. All the subjects provided written informed consent before entering the study, which was conducted under the Declaration of Helsinki. The study was not advertised, and no remuneration was offered.

The inclusion criteria were as follows:Presence of prolapse (cystocele and/or hysterocele and/or rectocele) ≥ stage 2 POP-Q;Indication for vaginal surgery (vaginal hysterectomy in case of hysterocele; anterior and posterior fascial repair for cystocele and rectocele, respectively).

### 2.1. Exclusion Criterion was Previous Vaginal Surgery

Of the 25 women recruited, 5 dropped out. Two women dropped out because of anxiety during the MRI examination, one did not give any reason, and two rejected postoperative MR. Twenty women completed the study. Women underwent preoperative clinical examination by the same doctor (MGM) to collect history and evaluate urinary symptoms and stage prolapse through the POP-Q system; women, also, underwent urodynamic examination. Then, they underwent an MRI performed by a radiologist who did not know the clinical staging of the prolapse. Urogynecological examination and MRI were repeated 3 months after vaginal surgery for pelvic organ prolapse.

### 2.2. MRI Imaging Protocol

MR examinations were performed with a closed-configuration superconducting unit with 1.5 T field strength (GESigna HDx 1.5 T, GE Medical Systems, Milwaukee, WI, USA) with 57.2 mT/m gradient strength and 120 T/m/s slew rate, by using an 8-channel high-resolution torso coil with array spatial sensitivity technique (ASSET) parallel acquisition.

### 2.3. Preparation

Patient preparation and cooperation were crucial for the success of the study. Before the examination, patients were administered an enema and provided instructions on the maneuvers to be performed inside the magnet. Patients were encouraged to wear a large pad, a strategy that served the dual purpose of preventing the soiling of the MR bed and minimizing psychological discomfort. Additionally, it was determined that the bladder should be filled to approximately half capacity. Inside the gantry, the rectum was distended with approximately 120 mL of ultrasound gel (hyperintense on T2 and FIESTA sequences) introduced through a Nelaton catheter (20 Ch, 6.67 mm × 360 mm) (Bicakcilar, Istanbul, Turkey) and a 50 mL catheter-tip syringe. The degree of straining was monitored using a respiratory gating device positioned around the patient’s waist. Inside the gantry, the patient lay supine, with their feet first and knees slightly flexed. This positioning facilitated the evacuation of the rectal contrast agent during defecation.

### 2.4. Sequences

Our protocol included the acquisition of the following:-High-spatial-resolution static sequences for studying the morphology of the levator ani;-Dynamic sequences used to assess abnormalities of the pelvic organs during contraction, at rest, during straining, and during defecation.

Static sequences included T2-weighted fast spin-echo (FSE) sequences in the sagittal, axial, and coronal planes. The technical parameters for this sequence were time to repetition (TR)/time to echo (TE), 4675/100; flip angle, 90°; section thickness, 4 mm; interslice gap, 1 mm; bandwidth, 41.67 kHz; field of view (FOV), 32 cm; matrix, 320 × 224; several averages, 4; number of images, 26; acquisition time, 3 min 49 s.

Dynamic sequences were performed in the midsagittal plane identified on the T2-weighted FSE static images, with the pubic symphysis, urethra, vagina, rectum, and coccyx included in the FOV. In the dynamic phase, two types of sequences were used: T2-weighted single-shot fast spin-echo (SSFSE) and fast imaging employing steady-state acquisition (FIESTA) sequences acquired with the following parameters:-SSFSE (TR/TE, 708/90; flip angle, 90°; section thickness, 8 mm; bandwidth, 83.3 kHz; FOV, 34 cm; matrix, 384 × 224; several averages, 0.5; acquisition time for each image, 0.3 s) in the midsagittal plane, with sequential acquisition during contraction, rest, and straining;-FIESTA (TR/TE, 3.3/1.4; flip angle, 45°; section thickness, 8 mm; bandwidth, 125 kHz; FOV, 35 cm; matrix, 224 × 224; a number of averages, 1; number of images, 20; acquisition time, 20 s) in the midsagittal plane, with continuous multiphase acquisition during contraction, rest, straining, and defecation.

The overall examination time, including patient preparation, was approximately 40 min.

The parameters observed under MR were as follows:Pubococcygeal line (PCL): a line drawn from the inferior margin of the pubic symphysis to the last coccygeal joint [16];Radiological evaluation of prolapse: Prolapse severity can be easily graded according to the ‘‘rule of three’’: prolapse of an organ ≤ 3 cm below the PCL is mild, 3–6 cm below the PCL is moderate, and >6 cm below the PCL is severe [17,18].Levator plate angle (LPA): enclosed between the levator plate and the PCL where the ileo-coccygeus touches the coccyx. It is usually 11.7° ± 4.8 in healthy women under the Valsalva maneuver [16].Posterior vesicoureteral angle (VUA): the angle between the axis of the urethra and the bladder base. In normal conditions, it is about 90°.The H-line: extends from the inferior aspect of the pubic symphysis to the anorectal junction, represents the genital hiatus: on effort, it is 5.8 cm ± 0.5 [19].The M-line: dropped as a perpendicular line from the PCL to the posterior aspect of the H-line: normally on effort, it is 1.3 cm ± 0.5 [19].The total vaginal length (TVL): distance from the posterior part of the introitus to the proximal vagina; if the cervix is present, it corresponds to the posterior fornix; if the woman had a hysterectomy, it corresponds to the vaginal vault.Funneling: the opening of the proximal urethra on effort.The thickness of the mid urethra.

### 2.5. Surgery

Surgery for pelvic organ prolapse consisted of vaginal hysterectomy in cases of hysterocele and anterior and posterior fascial repair for the anterior compartment and posterior compartment, respectively. Women were given spinal anesthesia and antibiotic prophylaxis and then placed in the lithotomy position. After vaginal hysterectomy, suspension of the vaginal apex to the uterosacral ligaments and McCall culdoplasty (with uterosacral ligaments incorporated into the closure of the peritoneum and upper vagina) were performed for the prevention of vaginal vault prolapse. Surgical treatment of cystocele is based on the reconstruction of the anterior vaginal wall and anterior colporrhaphy, which is performed by placing an interrupted suture with Vicryl 2-0 that plicates the weakened fascial tissues. Surgical treatment of rectocele can be approached either transvaginally or transperineally and can be repaired with native tissue, with the plication of rectovaginal fascia in cases of central and lateral defects. We used native tissue for pelvic floor repair according to the FDA Safety Communication of 2011 (Update on Serious Complications Associated with Transvaginal Placement of Surgical Mesh for Pelvic Organ Prolapse) [20].

Descriptive qualitative and quantitative statistical analysis was performed using the Kolmogorov–Smirnov test. The comparison between preoperative and postoperative data was made using a concordance correlation coefficient for qualitative parameters and a paired Student’s T-test for quantitative parameters. The Wilcoxon signed-rank test was used to compare related samples. Statistical significance was for *p* < 0.05.

## 3. Results

Of the 20 women (median age 60), 12 underwent vaginal hysterectomy and anterior and posterior pelvic floor repair and 8 underwent vaginal hysterectomy and anterior pelvic floor repair. Fifty percent complained of urgency, 50% complained of stress urinary incontinence (SUI), and 100% complained of heaviness (Table 1). Table 2 reports the staging of prolapse for each compartment in women recruited according to clinical examination and magnetic resonance evaluation before surgery. We included women with clinical stage ≥ 2 POP-Q because otherwise there was no indication for surgery (Table 2). At the 3-month follow-up, women reported an improvement in urinary symptoms, no complaints of heaviness, and a reduction in urgency (40% vs. 50%) and SUI (20% vs. 50%). From the analysis of Table 2 and Table 3, we can observe that preoperative clinical examination tends to overestimate the degree of prolapse in comparison to MR; otherwise, postoperative clinical examination tends to underestimate the degree of prolapse. The postoperative evaluation through MR gave few differences, as shown in Table 4, between the two staging techniques (Table 3) (Figure 1). Moreover, MR examination showed hypotrophy of the puborectalis muscle in 12 women and hypotrophy of the ileo-coccygeus muscle in 14 women. One woman had damage and loss of convexity of the levator ani. Qualitative analysis between pre- and postsurgery showed no statistically significant difference in all the parameters considered at rest (Table 4).On the contrary, under the Valsalva maneuver, we found some significant differences: the PCL was reduced (*p* < 0.05), the H-line was 7 mm shorter (*p* < 0.05), the M-line was 5 mm shorter (*p* < 0.05), and the LPA was 6° smaller (*p* < 0.05). The VUA was 57° wider, but the difference was not statistically significant (Table 5) (Figure 2).

## 4. Discussion

While magnetic resonance imaging (MRI) is not mandatory for diagnosis, there is evidence suggesting that MRI can significantly contribute to more accurately guiding the selection of surgical techniques in 41.6% to 75% of patients with various forms of pelvic floor dysfunction [19,20,21,22]. From our data, the comparison between the clinical and radiological evaluation of prolapse showed some discrepancies both pre- and postoperatively, especially for the first and second stages. MR performs staging through objective measurements, so it is more accurate. We think that clinicians can exhibit the same bias in staging if they admit women to vaginal surgery based on symptoms and desire for improvement, not only on prolapse quantification. Postoperatively, we had improvement in the questionnaire on quality of life, which is the real objective of surgery for prolapse. Furthermore, the routine use of preoperative pelvic MRI seems to lack cost-effectiveness when employed to identify women with pelvic organ prolapse. Wyman et al. demonstrated that MRI may become cost-effective when used to identify women at high risk of surgical failure following apical repair for POP. Repeat surgery for pelvic organ prolapse recurrence leads to increased direct and indirect healthcare costs [23].

For this reason, we agree with the literature, which considers MR not required to make a diagnosis of prolapse even if this method is more accurate and could be indicated in complex pelvic floor disorders. Many studies were performed using this imaging technique to better visualize pelvic structures. In the literature, MR findings were compared to other imaging techniques in the study of the pelvic floor: Foti et al. [24] found that in outlet obstruction syndrome, MR imaging can provide a morphological and functional study of pelvic floor structures and may offer an imaging tool complementary to conventional defecography in the multicompartment evaluation of the pelvis. Other functional studies [25] using MR have been published on continence mechanisms. They demonstrate that during a cough, normal pelvic floor muscle function results in timely compression of the pelvic floor, providing additional external support to the urethra, thus reducing displacement, velocity, and acceleration. In women with stress urinary incontinence (SUI), who have weaker urethral attachments, this shortening contraction does not occur. Consequently, the urethras of women with SUI move further and faster for a longer duration. Our radiologic data, besides, showed a postoperative reduction in the PCL, H-line, M-line, and LPA at rest, which was not statistically significant. On the contrary, this reduction was significant under the Valsalva maneuver, which demonstrated the anatomical changes caused by surgical treatment, which correlate with improvement in symptomatology. The VUA, the angle between the axis of the urethra and the bladder base, is usually a rectal angle at rest; during micturition, it becomes opened and disappears to permit the passage of urine. The disappearance of this angle, as a consequence of surgery, causes a permanent funneling of the bladder neck and the proximal urethra which could cause de novo incontinence. In 1952, Roberts [26] and Jeffcoate [27,28] reported a series of observations on the posterior urethrovesical angle and its relationship with urinary continence in females. These authors used the technique of lateral cystourethrography as their main method of observation: they concluded that urinary continence depends upon a posterior urethrovesical angle of about 100 degrees which is maintained by the intrinsic musculature of the bladder neck. Stress incontinence is characterized by loss of the posterior urethrovesical angle that can rarely be restored by anterior colporrhaphy. Dutton also found a significant correlation between the absence of the posterior urethrovesical angle and stress incontinence (91.5%), but only a moderate correlation between the presence of the angle and urinary continence (66.7%). It was concluded that the posterior urethrovesical angle is likely not the sole determinant of urinary control, but rather, it is closely linked to other contributing factors [29].

We found that the VUA, in a supine position before surgery for prolapse, was about 156° at rest. After the correction of the cystocele, it was wider, and this difference was not statistically significant either at rest or on effort. Moreover, our data reported a significant improvement in SUI (50% pre versus 20% post). We found that the widening of VUA after correction of cystocele is not statistically significant and did not alter the static or dynamic balance of pelvic organs. The limits of our study are the small number of women recruited and the lack of comparison with a different treatment of prolapse. These are preliminary data useful for understanding whether clinicians and radiologists can speak a common language about prolapse and understanding the potentiality of MRI as a diagnostic technique. Some larger studies should be performed to better understand whether MRI can predict recurrence.

## 5. Conclusions

Based on our data, static magnetic resonance imaging does not add any further information on the management and efficacy of surgical treatment for prolapse. Dynamic magnetic resonance imaging can be useful for further investigating the balance and functionality of pelvic organs, which can be compromised by prolapse. Moreover, radiologic measurements demonstrated that vaginal correction of prolapse is efficient and does not alter urethral mobility and continence mechanisms.

## Figures and Tables

**Figure 1 medicina-59-02074-f001:**
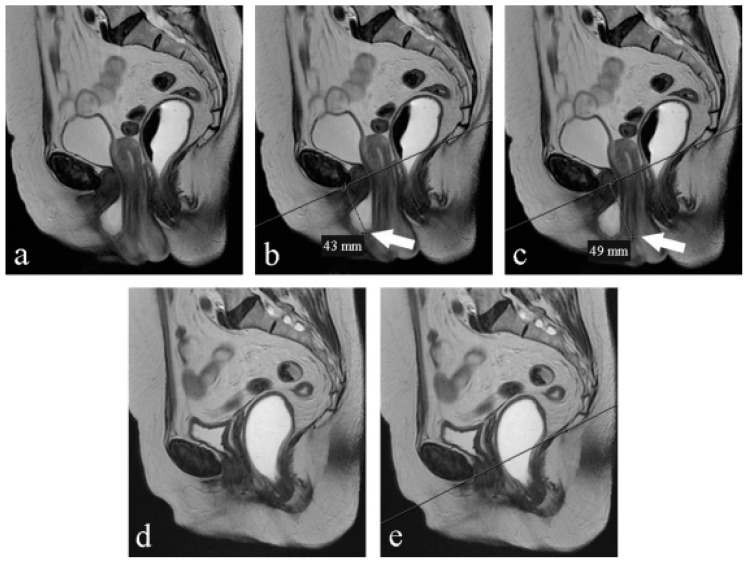
(**a**–**e**) Midsagittal T2-weighted FSE MR images, obtained at rest: preoperative (**a**–**c**) and postoperative (**d**,**e**) evaluation. (**a**) The bladder base and the uterine cervix descent below the PCL; (**b**) the bladder base (arrow) descends 4 cm below the PCL (line): stage 2 cystocele; (**c**) the uterine cervix (arrow) descends 5 cm below the PCL (line): stage 2 hysterocele; (**d**) cystocele and apex prolapse are no longer visible; (**e**) at rest, the bladder base and vaginal vault are located above the PCL (line).

**Figure 2 medicina-59-02074-f002:**
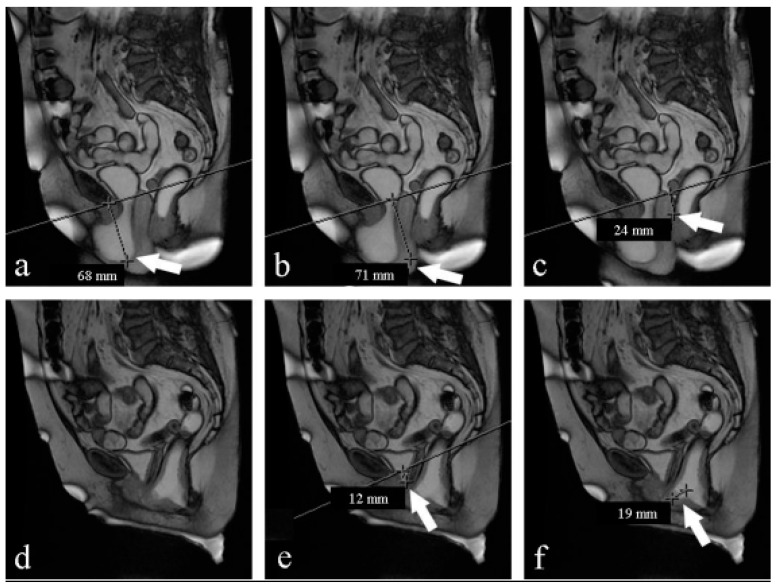
(**a**–**f**) Midsagittal MR FIESTA images, obtained under Valsalva maneuver: preoperative (**a**–**c**) and postoperative (**d**–**f**) evaluation. (**a**) Bladder base (arrow) descent >6 cm under PCL (line): stage 3 cystocele; (**b**) uterine cervix (arrow) descent >7 cm under PCL (line): stage 3 hysterocele; (**c**) Douglas (arrow) descent 2.4 cm under PCL (line): stage 1 elitrocele; (**d**) absence of prolapse; (**e**) bladder base (arrow) descent 1.2 cm under PCL (line): stage 1 cystocele; (**f**) anterior rectocele (arrow) <2 cm: stage 1 rectocele.

**Table 1 medicina-59-02074-t001:** General data and urinary symptoms.

General Characteristics	
Age	60 ± 7
Menopause	90%
Vaginal delivery	2.5 (average)
Forceps or vacuum	40%
Macrosoma	40%
Preoperative symptoms	
Heaviness	100%
SUI	50%
Urgency	50%
Frequency	30%
Retention	70%

**Table 2 medicina-59-02074-t002:** Staging of prolapse preoperatively.

GRADE	2		3		4	
	Clinic	MR	Clinic	MR	Clinic	MR
Cystocele	50%	70%	50%	30%	0%	0%
Hysterocele or apex	40%	80%	10%	10%	30%	0%
Rectocele	10%	10%	0%	0%	0%	0%

Percentage of presence of each stage of prolapse in women recruited before surgery for prolapse.

**Table 3 medicina-59-02074-t003:** Staging of prolapse postoperatively.

GRADE	0		1		2		3		4	
	Clinic	MR	Clinic	MR	Clinic	MR	Clinic	MR	Clinic	MR
Cystocele	**80%**	**40%**	**10%**	**40%**	10%	20%	0%	0%	0%	0%
Hysterocele or apex	**100%**	**80%**	**0%**	**20%**	0%	0%	0%	0%	0%	0%
Rectocele	**70%**	**30%**	**20%**	**40%**	10%	30%	0%	0%	0%	0%

Percentage of each stage of prolapse at three months after surgery for prolapse. Bold font indicates the stages of POP for which we consider the prolapse cured.

**Table 4 medicina-59-02074-t004:** Pre- and postoperative measurements at rest.

At Rest	Pre Average ± DS	Post Average ± DS	*p*
PCL (mm)	100 ± 12.22	100.2 ± 12.44	0.76
H-line (mm)	58.4 ± 8.96	57.4 ± 6.69	0.43
M-line (mm)	55.9 ± 8.03	52.3 ± 6.67	0.06
TVL (mm)	66.3 ± 26.33	59.4 ± 21.67	0.27
Urethra (mm)	5.1 ± 0.57	5.2 ± 0.41	0.55
LPA (°)	22.6 ± 9.05	21.2 ± 8.52	0.38
VUA (°)	156.7 ± 48.05	161 ± 11.67	0.77

PCL: pubo-coccigeal line. H-line: the line from the inferior aspect of the pubic symphysis to the anorectal junction. M-line: perpendicular line from the PCL to the posterior aspect of the H-line. TVL: the total vaginal length. LPA: the levator plate angle. VUA: posterior vesicourethral angle

**Table 5 medicina-59-02074-t005:** Pre- and postoperative RM measurements under Valsalva maneuver.

Valsalva	Pre Average ± DS	Post Average ± DS	*p*
PCL (mm)	103.3 ± 11.26	102.6 ± 11.96	0
H-line (mm)	82.2 ± 11.75	77.6 ± 12.75	0.008
M-line (mm)	69.8 ± 6.42	65 ± 9.64	0.001
LPA (°)	50.7 ± 9.5	44.7 ± 11.12	0.001
VUA (°)	130 ± 68	187.4 ± 53.32	0.59

PCL: pubo-coccigeal line. H-line: the line from the inferior aspect of the pubic symphysis to the anorectal junction. M-line: perpendicular line from the PCL to the posterior aspect of the H-line. LPA: the levator plate angle. VUA: posterior vesicourethral angle.

## Data Availability

Data sharing is not applicable to this article.

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
