# Peer review of "Role of Magnetic Resonance Imaging in Pelvic Organ Prolapse Evaluation"

_medicina, 2023, doi:10.3390/medicina59122074_

Round 1

Reviewer 1 Report

Comments and Suggestions for Authors

This study is interesting and well orgnized. I have some suggstions to improve the writing of manuscript.

This retrospective study was conducted between October 2018 and February 2019. Please clarify how the authors decided on the duration of the study.

Figures and Tables

Did the authros check the Figures and Tables before submission? Please delete red wavy lines.

Discussion

This is a run-on paragraph and is there are no line breaks. Long syntaxes tend to confuse readers and may lead to misinterpretation. Short syntaxes are preferred for better clarity and readability. 

Comments on the Quality of English Language

I recommend that the manuscript be reviewed by a person with professional proficiency in English to correct errors in grammar, punctuation, word choice, and sentence construction to improve the flow of ideas expressed in the article to ensure that the document reads as though written by a native English speaker.

Author Response

Best regards

Reviewer 2 Report

Comments and Suggestions for Authors

1.  Please discuss how MRI evaluation contributes to the choice of surgical methods and postoperative management etc of pelvic floor dysfunction.

2. Please discuss the cost-effectiveness of MRI evaluation (compared with conventional diagnosis) in the clinical practice of pelvic floor dysfunction.

3. Please present not only p values but also the odds ratio and 95% confidence intervals for each statistical comparison.

Author Response

Best regards
